# Relative income and its relationship with mental health in UK employees: A systematic review

**Bethany Croak**[1,2*], **Laura E. Grover**[2], **Simon Wessely**[2], **Kalpa Kharicha**[3], **Danielle Lamb**[4‡], **Sharon A.M. Stevelink**[1,2‡]

1 Department of Psychological Medicine, King's College London, London, United Kingdom, 2 King's Centre for Military Health Research, King's College London, London, United Kingdom, 3 Health & Social Care Workforce Research Unit, The Policy Institute, King's College London, London, United Kingdom, 4 Department of Primary Care and Population Health, UCL, London, United Kingdom

‡ These authors are joint senior authors.
* bethany.croak@kcl.ac.uk

## Abstract

***Purpose:*** The relative income hypothesis theorises that one's earnings relative to others exert a greater influence on subjective wellbeing than absolute income. Understanding the relationship between relative income and mental health could contribute to employee wellbeing. This review aimed to summarise the defining features of relative income in relation to mental health and how it is measured in the literature. In addition, it aimed to explore the relationship between relative income and mental health in those currently employed in the UK. ***Methods:*** Nine electronic databases were searched using a pre-defined search strategy: PubMed (including MEDLINE and PubMed Central), PsycINFO, Scopus, Web of Science, Global Health, JSTOR, Business Source Complete (EBSCO), ScienceDirect and Emerald. The protocol was pre-registered on PROSPERO (CRD42023408657). Quantitative and qualitative studies and grey literature, which described the defining features and measurement of relative income and its impact on mental health among UK employees, were included. ***Results:*** After screening, 13 studies were included in the review. A conceptualisation of relative income revealed that an income comparison is either researcher-defined using averages or self-assessed based on a person's perception. Having a lower income than the reference group was commonly associated with diminished wellbeing, though moderating factors (gender, income inequality and composition of reference group) were identified. ***Conclusions:*** Having a lower income than the reference group is associated with poorer wellbeing. Implications for practice and policy are considered amidst the UK's 'cost of living crisis' and ongoing pay disputes in various sectors.

## Introduction

Can money buy happiness? This debate has long-fuelled dinner parties and academic discussions alike. A core catalyst of this debate was Easterlin's paper [1] and the subsequently coined

**Data availability statement:** All relevant data are within the paper and its Supporting Information files.

**Funding:** This work is part of a PhD nested within the NHS CHECK study and funded by the ESRC [ES/P000703/1]. This report is independent research supported by the National Institute for Health and Care Research ARC North Thames. The views expressed in this publication are those of the author[s] and not necessarily those of the National Institute for Health and Care Research or the Department of Health and Social Care. LEG is funded by the Armed Services Trauma Rehabilitation Outcome [ADVANCE] Charity. Key contributors to this charity are the Headley Court Charity [principal funder], HM Treasury [LIBOR Grant], Help for Heroes, Nuffield Trust for the Forces of the Crown, Forces in Mind Trust, National Lottery Community Fund, Blesma - The Limbless Veterans and the UK Ministry of Defence.

**Competing interests:** BC, LEG and DL have no competing interests. SW is a non-executive director at NHS-England. KK is the deputy director and part funded by the NIHR Policy Research Unit in Health and Social Care Workforce. SAMS is supported by the National Institute for Health and Care Research [NIHR], Maudsley Biomedical Research Centre at South London and Maudsley NHS Foundation Trust and the National Institute for Health and Care Research NIHR Advanced Fellowship [Dr Sharon Stevelink NIHR300592]. The views expressed in this publication are those of the authors and not necessarily those of the ESRC, the NHS or the NIHR.

phrase 'The Easterlin Paradox', describing the contrasting relationship between income and happiness on the individual and macro level. On an individual level, wellbeing was strongly positively associated with income rises. However, on a macro level, a country's income growth (often measured by Gross Domestic Product (GDP)) over time did not correspond to rises in average population levels of happiness.

Many have attempted to address this paradox, and a widely cited theory is the 'Relative Income Hypothesis'[2], which states that an individual's attitude to saving and consumption is influenced more by their income in relation to others than by their actual income in its own right [3]. The application of this theory has been expanded beyond saving and consumption behaviour to utility (i.e., happiness, satisfaction, or pleasure) or what in the broader literature has come to be known as subjective wellbeing (SWB). Many have attempted to study the relative income effect in various scenarios. In a sample of 5000 British people, relative income was more strongly correlated with SWB than absolute income; a change in income from one standard deviation below the mean wage to one standard deviation above the mean led to an increase of 1.6 points in SWB [4]. On a more granular level, a study of married couples in Sweden showed that an increase in the wife's salary beyond that of her husband increases the husband's likelihood of a mental health diagnosis by 8%–11% [5].

SWB captures a wide range of factors, including physical health, satisfaction with one's job and home life, personal safety, and mental health [6]. There is a strong socioeconomic gradient in SWB, such as physical health, as demonstrated by Marmot's seminal Whitehall Study [7]. Whilst many domains of satisfaction, such as job, family, and life satisfaction, have been the focus of relative income research, mental disorders specifically have been neglected somewhat in the literature. Poor mental health is the leading cause of disability worldwide [8] and the relationship between poverty and mental health is well established; for example, those in the lowest 20% income bracket in Great Britain are two to three times more likely to develop mental health problems [9]. However, income and wealth are meaningful to all, not just those in poverty. To our knowledge, no previous work has summarised the literature on relative income and its relationship with wellbeing and mental health in employed persons in the UK.

In addition to the burden poor mental health can put on health services and the economy, poor satisfaction with pay, working conditions and job security can have implications for industries and public services. In the UK, pay and income have become a particularly pertinent subject amid the 'cost of living crisis' [10] and ongoing industrial action across several sectors, such as the National Health Service (NHS) [11] and rail companies [12]. Policymakers and corporations tend to approach these negotiations through the lens of absolute pay with policies such as the National Living Wage [13]. However, unions and employees focus on pay changes over time and pay in relation to other employees and industries [14]. Therefore, understanding relative income and its impact on wellbeing may help develop more informed policies such as pay deals for public sector workers and taxation thresholds.

This review aims to summarise the defining features of relative income in relation to mental health and how it is measured in the literature. In addition, it seeks to explore the relationship between relative income and mental health in those currently employed in work in the UK. Mental health is defined in this review to encapsulate mental disorders and subjective wellbeing (including life satisfaction).

## Method

### Design

This systematic review was conducted following Cochrane methodology and Preferred Reporting Items for Systematic reviews and Meta-Analyses (PRISMA) guidelines (S1 File).

Prior to commencing the review, the protocol was registered with the international prospective register of systematic reviews (PROSPERO) (CRD42023408657).

## Search strategy

Nine electronic databases were searched in February 2023 and updated in March 2024: PubMed (including MEDLINE and PubMed Central), PsycINFO, Scopus, Web of Science, Global Health, JSTOR, Business Source Complete (EBSCO), ScienceDirect and Emerald. All databases were searched using pre-defined terms related to (1) relative income, (2) mental health, and (3) working populations. See S2 File for the full search strategy. In addition, grey literature was searched using the OpenGrey database and key organisations' websites.

In addition, PROSPERO was searched to identify planned or ongoing systematic reviews and meta-analyses of relevance; no duplicate reviews were identified. Further, backwards and forwards citation checking of the included studies was used. An expert group was consulted to identify further literature, consisting of academics whose research focuses on pay and reward, employment, economics and health inequalities.

## Study selection criteria

The search included all original, peer-reviewed work and high-quality grey literature published in English that captured both or either: i) a conceptualisation or clear definition of relative income; and ii) an exploration of the relationship between relative income and mental health. Restrictions were placed on the population only to include studies of those in employment in the UK (referred to as UK employees) and those over 18 years of age. In the UK, individuals can be employed from 16 years old, but by law, they still need to be in training (apprenticeship) or education until they are 18 [13]. As such, it was deemed appropriate to restrict the search to studies which included those over 18 years of age. Conference proceedings, case studies, editorials, systematic reviews, book chapters and PhD dissertations were excluded.

## Screening and data extraction

Following an initial search, all identified studies were captured in EndNote 20 [15], and duplicates were removed. BC independently reviewed the titles and abstracts of all studies. The full text of studies deemed relevant were then screened. A second reviewer (LEG) independently reviewed 10% of the search results at each screening stage to check for interrater reliability.

The reviewers (BC and LEG) independently decided which studies met the eligibility criteria to be included in the review and, at the full-text screening stage, noted any reasons for exclusion. Any discrepancies were resolved through discussion. Interrater reliability was 94% at the title/abstract and 100% at the full-text screening stage. For a full list of identified studies and screening decisions, please see the S4 File.

The following data were extracted independently by BC for all included studies, where available: general information (e.g., title) and study characteristics (e.g., design, sample size). The definitions of relative income and how it was measured were summarised and informed the conceptualisation of relative income. There was a large amount of heterogeneity in the statistical analysis used for the relative income effect on mental health. The coefficients reported were not meaningful on their own as the marginal effects of regressors often needed to be interpreted (e.g., with a probit regression). However, where more intuitive interpretations are provided, such as percentage increases, these are reported. In addition, this review included both quantitative and qualitative literature. Therefore, it was not considered appropriate to do a meta-analysis or report the coefficients in this review but to summarise the main

conclusions (qualitative themes and regressor effects of relative income) of the literature using a narrative synthesis [16]. For the full data extraction table, please see S5 File.

### Quality assessment

BC carried out a quality assessment of each included study using the National Heart, Lung, and Blood Institute (NHLBI) checklist [17] for quantitative studies and grey literature, and the CASP (Qualitative) Checklist [18] for qualitative studies. Both of these checklists are designed to help the reviewer focus on the key concepts for evaluating the internal validity of a study. In regard to the rating, reviewers are discouraged from tallying up the checklist to produce a rating. Therefore, BC gave a rating of 'poor' 'fair' or 'good' based on their judgement of the risk of bias for each included study. LEG carried out an independent quality assessment on 50% of the included studies, and any discrepancies were discussed until an agreement was reached. Studies were not excluded based on quality assessment results, but the assessment provided an understanding of the quality of research in the field. The full quality assessment spreadsheet can be found in the S6 File.

## Results

Overall, 4506 studies were identified, of which 936 duplicates were removed (Fig 1). The PRISMA diagram includes the original search and the update. The title/abstracts of 3570 studies were screened, of which 199 were retained for full-text screening. Experts and cross-referencing identified no additional studies. A further paper was identified through a grey

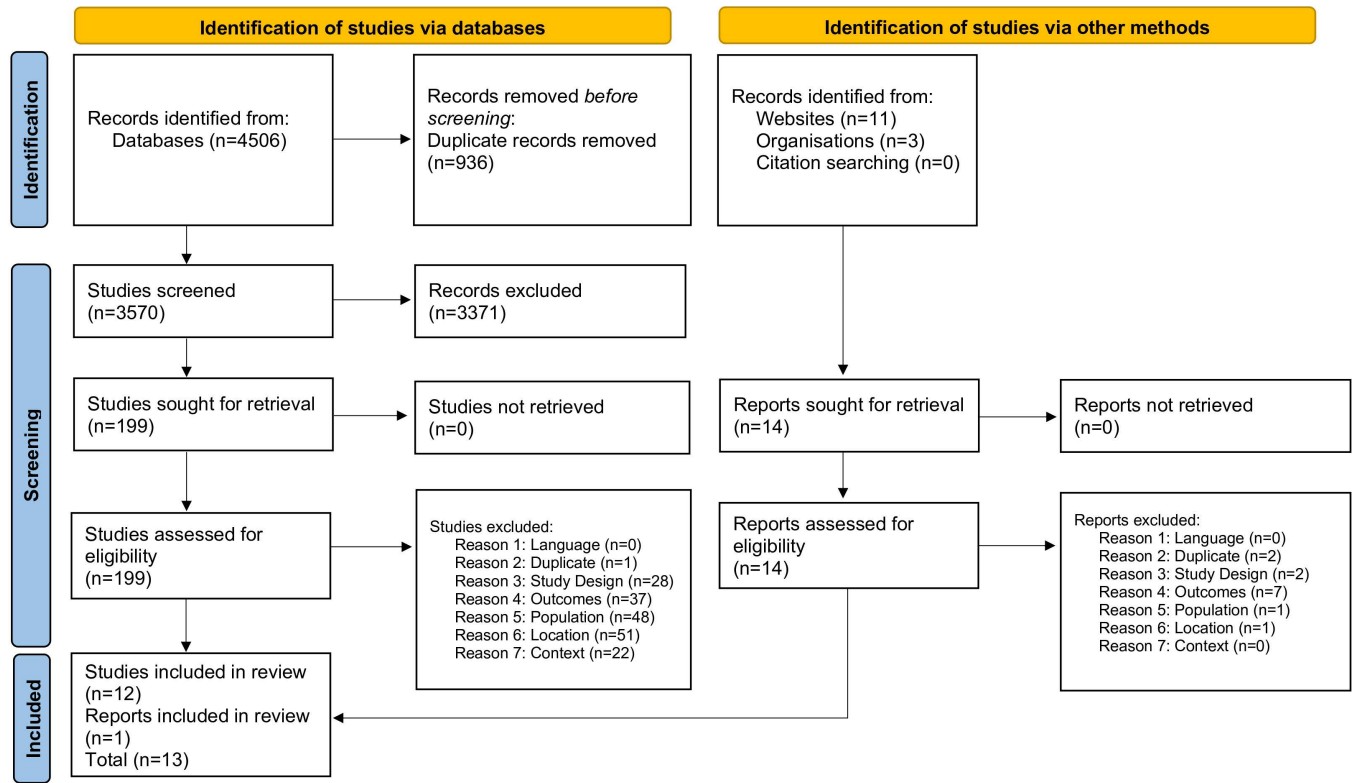

**Fig 1. PRISMA flowchart.**

literature search. Overall, 13 studies met the inclusion criteria (S3 File) and are discussed in detail in this review.

## Study characteristics

Included studies had a range of study designs; the majority were longitudinal studies (n = 6) [19–24]. Other designs included cross-sectional studies (n = 4) [25–28], a randomised controlled trial [n = 1] [29], a qualitative study using focus groups (n = 1) [30] and grey literature (n = 1) [31].

The studies often explored multiple exposures, such as income rank (position within an income distribution [32]), income inequality (how unevenly income is distributed within a population), and relative income. Table 1 lists only the main exposure of interest: relative income. A more detailed summary of how this has been conceptualised and measured is provided below in section 3.3. Similarly, only the data source and designs relevant to the aim of the review are described in Table 1.

## Quality assessment

Of the 13 studies included in the review, most studies received a quality assessment score of 'good' (n = 8) [19–24,30,31] with some scoring fair (n = 4) [25,27–29] and one study scoring 'poor' [26]. Common reasons for scoring 'fair' in the quality assessment included the absence of a sample size justification or power calculation and, as many were cross-sectional, the exposure not being assessed prior to the outcome being measured. A rating of poor was given if the reviewer (BC) deemed there to be a significant flaw; these reasons included having a poorly defined aim or not describing the sample sufficiently. The full quality assessment spreadsheet can be found in the S6 File.

## Conceptualisation of relative income

Relative income was broadly understood by all included studies to be one's income in comparison to another individual or group's income (known as the reference group).

Methods of measurement were divided into researcher-led and self-assessed. For the researcher-led methods, the most common way of calculating how someone compared was to use a binary measure of an individual being above or below the group mean income [20–22]. Socioeconomic status was used in one paper as a proxy for relative income; status increases as the individual outperforms the mean income achieved amongst their reference group [28]. Similarly, ratio was used, for example, the ratio of the individual's income to the region income per capita [26]. The median was also used as a proxy, for example by the UK government, who defined relative low income as, "an individual is in relative low income (or relative poverty) if they are living in a household with income below 60% of median household income in that year" [31].

Self-assessed measurement was done by asking participants how they perceived their income in relation to the reference group. For example, one study asked participants the following question in a survey: "Compared to the financial situation of your neighbours/most of your friends/work colleagues, would you say your household is…" with responses on a 5-point Likert scale ranging from "much better off" to "much worse off" [23]. A second study used a similar approach and asked participants to self-assess their relative income status compared to "individuals of similar professional standing and characteristics" [27].

In addition, the current review identified qualitative descriptions of relative status. In the study by Davidson et al. [30], reference groups in the context of socioeconomic status

**Table 1.** Characteristics of included studies (n = 13).

| Paper | Design | Data source | Year/waves | Total sample (n) or observations* | Exposure of interest | Reference Group (RG) | Outcome |
|---|---|---|---|---|---|---|---|
| Becchetti et al. [19] | Longitudinal. Individuals are followed over 14 waves. | British Household Panel Survey (BHPS). | Waves 1–14: 1991–2005. | n = approximately 10000 (71228 observations). | 1. Relative personal income (mean). 2. Relative job income (mean). | 1. Peer groups based on gender, education, location and age cohorts. 2. Peer groups based on gender, age cohorts and working environment. | Life satisfaction: Participants are asked to evaluate their overall level of life satisfaction on a 1–7 scale. |
| Becchetti et al. [25] | Repeated cross-sectional. Cross-sectional data at the individual level over a period of seven waves. Individuals are not followed across different waves. | European Social Survey (ESS). | Waves 3–9: 2006, 2008, 2010, 2012, 2014, 2016 and 2018. | Observations = 165270. | Relative income (mean). | Regional sample population. | Life satisfaction: Participants are asked "How much are you satisfied with your life as a whole?". The answers range from 0 (not satisfied at all) to 10 (really satisfied). |
| Blanchflower et al. [26] | Repeated cross-sectional. Cross-sectional data at the individual level for each year in the time period. Individuals are not followed over time. | The Eurobarometer Survey. | 1973–1998. | n = approximately 55000 | Relative income (ratio). | Regional population. | Life satisfaction: Participants are asked "on the whole, are you very satisfied, fairly satisfied, not very satisfied, or not at all satisfied with the life you lead?" |
| Brown et al. [24] | Longitudinal. Unbalanced panel. | Understanding Society Study. | Waves 1–3: 2009–2013. | n = 40335 (99430 observations). | Relative Income (mean). | Test two reference groups:1. RG was comprised of individuals with similar characteristics: age, education, and gender. 2. The RG is based on a spatial definition: the average in the local authority district. | Life satisfaction: "Please tick the number which you feel best describes how dissatisfied or satisfied you are with your life overall." Measured on a 7 point scale; 1 indicates "completely dissatisfied," 7 "completely satisfied." General Health Questionnaire (GHQ-12) [32] |
| Davidson et al. [30] | Qualitative focus groups | N/A | January 1999-Feburary 2000. | n = 76. | "Do they consider relative socioeconomic status to be important, and do they compare themselves to, or feel judged by others?" | Not specified. | Qualitative descriptions. |

*(Continued)*

Table 1. (Continued)

| Paper | Design | Data source | Year/waves | Total sample (n) or observations* | Exposure of interest | Reference Group (RG) | Outcome |
|---|---|---|---|---|---|---|---|
| FitzRoy et al. [20] | Longitudinal – panel data | BHPS and Understanding Society Study | BHPS Waves 6–18: 1996–2009 BHPS harmonised with Understanding Society Study, Waves 2–7: 2010–2017 | n = 27262 (207907 observations). | Relative income rank. Household income is used. | RGs based on similar characteristics: age, sex, education, region and survey wave. | Life satisfaction: exact question not specified. |
| FitzRoy et al. [21] | Longitudinal – panel data | BHPS | Waves 6–10 and 12–18: 1996–2008. | n = 25681 (153189 observations). | Household income measures real household income, using the Consumer Prices Index as deflator. Comparison income measures the average real household income within a reference group. | RGs defined based on similar characteristics: age, sex, education and region. | Life satisfaction: Self-reported life-satisfaction is measured on a 7-point scale, 1 being the lowest value, while 7 is reported by individuals who are very satisfied with their life overall. |
| Francis-Devine [31] | Report (grey literature). | Secondary data summarised – mental health information taken from Family Resources Survey and HBAI dataset (2018/19) and Understanding Society (2014/15–2017/18), SMC analysis. | Time period: 2014, 2018 and 2019. | N/A - Secondary data summarised. | Relative low pay: An individual is in relative low income [or relative poverty] if they are living in a household with income below 60% of median household income in that year. | UK population. | Mental health (validated self-report measure used: GHQ-12). |
| Fumagalli et al. [29] | Randomised Controlled Trial. | Added two questions (treatment or control) to wave 5 of the Understanding Society Study. | Wave 5: 2012. | n = 1224 households. | How dissatisfied or satisfied are you with your (domain)? | Treatment Group: RG is same gender. Control group: No RG is specified. | Satisfaction with 4 domains:<br>• Health<br>• Household income<br>• Amount of leisure time<br>• Life overall |
| Lorgelly et al. [22] | Longitudinal – panel data. | BHPS. | Waves 1–12: 1991–2002. | n = 8645. | Relative income - log of average income within each region. | Individuals in same region and survey year. | Self-rated health: Participants were asked "Compared to people of your own age, would you say that your health over the past 12 months has on the whole been excellent, good, fair, poor or very poor?" |
| Parker et al. [28] | Cross-sectional. | British Social Attitudes [BSA] Surveys. | 2004. | n = 1704. | Relative income – below or above mean of RG. | Similar occupational class (limited information on how this has been defined). | Family satisfaction: Participants were asked "All things considered, how satisfied are you with your family life". Participants are given a choice of seven ordered responses, ranging from completely unsatisfied to completely satisfied. |

*(Continued)*

**Table 1.** (Continued)

| Paper | Design | Data source | Year/waves | Total sample (n) or observations* | Exposure of interest | Reference Group (RG) | Outcome |
|---|---|---|---|---|---|---|---|
| Theodos-siou et al. [27] | Cross-sectional. | SOCIOLD research project. | 2004. | Not stated. | Relative income status – self-assessed [no further details provided]. | Individuals of similar professional standing. | Mobility score: participants were asked how difficult they found activities such as bathing and walking. Higher scores would reflect lower difficult in mobility so better physical health. Self-assessed health: Participants were asked to assess their own health on the whole over the last 12 months. Responses ranged from very bad (1) to very good (5). Mental health Participants were asked to comment on whether they have been feeling as if they are (a) enjoying they things they used to enjoy, (b) looking forward with enjoyment to things, (c) laughing and seeing the funny side of things, and [d] less irritable. Their answers ranged from much less than usual (1) to much more than usual (5). |
| Yu [23] | Longitudinal – panel data. | English Longitudinal Study of Ageing (ELSA). | Waves 2–5: 2004–2011. | Observations = 24502. | Self-perceived relative income: "Compared to the financial situation of your Neighbours/ most of your friends/ work colleagues, would you say your household is....." Respondents may choose an answer among i) "much worse off," ii) "a bit worse off," iii) "about the same," iv) "better off," and v) "much better off." | Three RGs: Friends, colleagues, and neighbours. | Life satisfaction: Participants were asked to report how much they agree or disagree with the following statement: "I am satisfied with my life." The choices included i) "strongly agree," ii) "agree," iii) "slightly agree," iv) "neither agree nor disagree," v) "slightly disagree," vi) "disagree," and vii) "strongly disagree." |

* For panel data, number of individuals per wave was not described. The n refers to the number of individuals included in the regression models and observations refers to the total sample size as opposed to individuals across the whole sample.

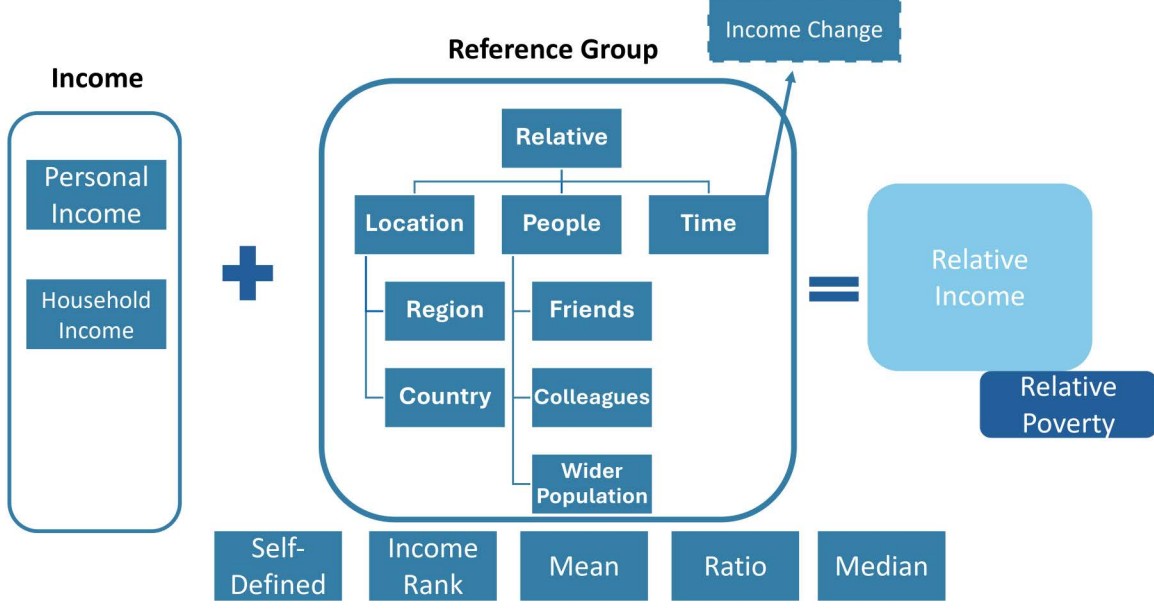

**Fig 2. Conceptual framework of relative income.**

emerged, represented by a "language of division" with participants using terms such as "us" and "them" when discussing different socioeconomic classes. Individuals in this study appeared to identify with one class or group of people and had an awareness of where they were placed in social hierarchies. Examples such as buying a branded loaf of bread compared to a supermarket-own bread were used by participants to distinguish different wealth levels.

The various measurements of relative income as well as the reference groups used were synthesised and a conceptual framework on relative income was developed (Fig 2).

## Relationship between relative income and mental health, wellbeing and satisfaction

The included studies reported a range of mental health and wellbeing measures (Table 1). The most widely used was life satisfaction (alone or in addition to another measure) (n = 10) [19–21,23–26,29]. The most common way of evaluating life satisfaction was not by a validated measure but by using a single question in which individuals are asked to rate how satisfied they are with their life on a scale of 1–10, but this did vary slightly between studies (see Table 1). Other types of satisfaction were used as an outcome, including family satisfaction (n = 1) [28], health satisfaction (n = 1) [29], income satisfaction (n = 1) [29] and satisfaction with the amount of leisure time (n = 1) [29]. Some studies (n = 2) asked individuals to rate their health (including mental health) [22,27]. One study asked individuals to rate their happiness [26]. One study [24] used a validated mental health measure, the general health questionnaire (GHQ-12), which is commonly used to examine psychological distress [33]. Similarly, a UK government report included mental health as an outcome; this was from secondary data taken from the Family Resources Survey and HBAI dataset (2018/19) and Understanding Society (2014/15–2017/18), which used the GHQ-12 too [31].

Two main effects were described in the literature; the comparison effect whereby individuals compared themselves to others and, if they did not earn as much or perceived themselves

to earn less then this was dissatisfactory (negative effect); and an information or aspirational effect (positive) in which individuals saw a higher income in the reference group as satisfactory, as they used the comparison income of the reference group to form expectations about their own future income [34]. From this point onwards, these two effects are noted as negative and positive, respectively.

Of the studies that quantitively assessed the relationship between relative income and satisfaction/mental health (n = 12), six studies found a negative effect (if an individual's income was below that of the reference group, wellbeing decreased) [19–23,25–27,31]. Five studies found mixed effects dependent on different factors (e.g., gender, reference group) [20,21,24,28,29] and one study found no effect [22]. The main conclusions and effect direction for the quantitative studies are summarised in Table 2, and the direction of effect and moderating factors (third variables which changed the magnitude or direction of the effect of relative income on mental health) are summarised in Fig 3.

Qualitative descriptions of emotions were reported by Davidson et al. [30] when participants were prompted to think about their relative socioeconomic status and its importance. Beyond some feelings of guilt about their relative privilege, there seemed to be no psychological consequence for those in the higher socioeconomic status groups. Conversely, those from lower socioeconomic groups described strong emotive reactions when considering their relative socioeconomic status including feelings of being judged by others, shame, or embarrassment. They also felt unheard and ignored by national and local governments.

## Income inequality

Income inequality is defined as the extent to which income is evenly distributed within a population (often measured by the Gini index). The literature described this as distinct to relative income and the two were listed as separate variables in analysis. However, income inequality was identified as a moderating variable in the relationship between relative income and mental health; Becchetti et al. [25] found negative relative income effects were stronger in areas where inequality was higher.

## Sub-groups

Some included studies looked at gender differences. In one study, women with income above the mean of their occupational class had a 53% increase in the probability of reporting themselves to be 'completely satisfied with family life', whereas no effect of relative income on family satisfaction was found in men [28]. Relatedly, gender differences were tested in a randomised controlled trial [29]. In a survey question, participants were randomly assigned to two conditions. Participants were prompted to evaluate their subjective wellbeing (measured as satisfaction with health, income, amount of leisure time and life overall) by comparing themselves with the same gender (treatment group) or to answer without reference to an explicit reference group (control). Increases in income and leisure satisfaction were found when women compared to women but no or little effect was found when comparing men in the treatment group to men in the control group. The authors interpret this as meaning women compare themselves to at least some men when unprompted and believe they are worse off than men. There was no difference in men's reported satisfaction when asked to compare themselves to other men or when the comparison group was not specified so the authors conclude men do not compare themselves to women. In addition, the authors also explored the impact of gender pay gaps; they found this effect in women was larger for those who worked in industries which had larger gender pay gaps [29].

In addition to gender, studies also stratified by age group. FitzRoy, Nolan [20] found a positive effect in those under 45 years old; they found that if the reference group had a higher

**Table 2. Quantitative results for the relationship between relative income and mental health in UK employees.**

| Paper | Outcome | Main effect | Magnitude estimates |
|-------|---------|-------------|---------------------|
| Becchetti et al. [19] | Life satisfaction | Negative | The decrease of personal income with respect to that of the peers is related to significant decreases in life satisfaction. |
| Becchetti et al.[ 25] | Life satisfaction | Negative | Negative relative income effects found and are relatively stronger when inequality is higher. |
| Blanchflower et al. [26] | Life satisfaction | Negative | Individuals compare to the richest percentile and the closer they get to this, the happier they get. |
| Brown et al. [24] | Life satisfaction and General Health Questionnaire (GHQ-12) | Mixed | The direction and significance of the relative income effect varies with reference group (RG) and estimation technique. For both life satisfaction and GHQ-12, when the RG is based on individual characteristics, significant negative relative income effects are apparent (if income is below the mean of the RG, life satisfaction and GHQ-12 values decrease) with the exception of the fixed effects estimates, where the effects are insignificant. When the RG is defined spatially, the estimated effects are all positive (happiness increases as RG income increases) but significance varies with estimation method. |
| FitzRoy et al. [20] | Life satisfaction | Mixed | In aggregate date: lower income than the reference group is associated with lower satisfaction. In disaggregated data: positive relative income effects for under 45 year olds and negative relative income effects for over 45 year olds. |
| FitzRoy et al. [21] | Life satisfaction | Mixed | Positive relative income effects for under 49 year olds and negative relative income effects for over 49 year olds. |
| Francis-Devine [31] | Mental health [secondary data, GHQ-12 used] | Negative | People in (absolute and relative) poverty are more likely to have poor physical and mental health, and low life and health satisfaction. |
| Fumagalli et al. [29] | Satisfaction with 4 domains: • Health • Household income • Amount of leisure time • Life overall | Mixed | When comparing themselves to other women on the four domains [listed in the second column], women report higher satisfaction. No effects between treatment and control for men. |
| Lorgelly et al. [22] | Self-rated health | None | No support for relative income hypothesis – relative income did not have an impact on health. |
| Parker et al. [28] | Family satisfaction | Mixed | In men: "status"* had no effect on family satisfaction. In women: Women with income above the mean of their occupational class had a 53% increase in the probability of reporting they were 'completely satisfied' with their family life. *status increases as the individual out-performs the income achieved amongst their reference group. |
| Theodossiou et al. [27] | Self-reported mental and physical health | Negative | An increase in the subjective social status assessment by one standard deviation point improved the mobility score [a measure of how well individuals could do everyday tasks like bathing and walking; higher scores indicate better physical health] by 7.5%, the self-assessed health by 22.4% and mental health by 14.4%. |
| Yu [23] | Life satisfaction | Negative | On average, individuals who perceive a bit lower or much lower income than their friends' rate their life satisfaction 0.60 and 1.3 points lower, respectively. The results show similar patterns when the reference groups are work colleagues and neighbours. The magnitude of the coefficients of income comparison against friends and neighbours are larger than that when colleagues is the reference group. For example, the negative impact on people's life satisfaction of perceiving much lower income than their friends' is around 0.5 points larger in absolute value than that of perceiving much lower income than colleagues'. |

income, they would have higher life satisfaction. FitzRoy et al. [21] found a similar effect with an age split of 49 years. This is coined the "tunnel effect", first introduced by Hirschman and Rothschild [35]. They suggested observing other people's faster progression or higher income can be positive if one interprets this as a sign that they will also achieve this income or

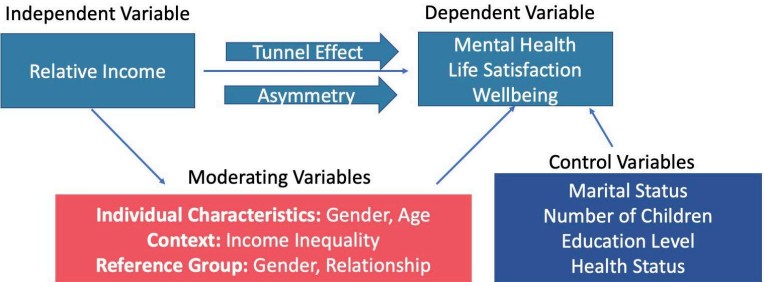

**Fig 3. Summary of review findings for the relationship between relative income and mental health.**

progression soon. They use the analogy of a tunnel: in a tunnel, a driver sees cars in the adjacent lane start to progress towards the exit while their lane is still immobile during a traffic jam.

## Asymmetry

An asymmetrical effect was found in that self-perceived relative income has an impact on life satisfaction but the decline in life satisfaction is much more significant due to perceiving a lower relative income in comparison to the rise in life satisfaction because of perceiving a higher relative income [23].

## Reference group

Various reference groups were used by studies (summarised in Table 1). Due to the heterogeneity of relative income definitions and statistical analyses, comparison of results by reference group across studies was not possible. However, two studies conducted such a comparison within their own analyses. Brown et al. [24] compared relative income effects when the reference group comprised of individuals with similar demographic characteristics such as age, education and gender and a spatial reference group which used the average income of those in your local authority district. They found negative relative income effects with the individual reference group and positive relative income effects with the spatial reference group (although these effects are less pronounced).

Yu [23] found negative relative income effects across all types of reference group: friends, neighbours and work colleagues. However, the magnitude of the coefficients when participants compared themselves to friends and neighbours were 0.5 points larger in absolute value than the coefficients found when comparisons were made to colleagues.

## Discussion

This is a conceptual review of 'relative income' followed by a systematic review exploring the relationship between relative income and mental health among UK employees. Although the literature on relative income and wellbeing is vast, a small number of studies met our inclusion criteria. In particular, few focused on individuals in the UK who were in paid employment. However, of the studies identified, most were of good quality.

### Summary of findings in relation to previous literature

Various definitions of relative income were used, but they broadly followed the same thread: it is the measure of one's income when compared to another person, or groups of individuals. This

was either researcher-led (e.g., individuals divided based on whether they were below or above the mean income of the reference group) or self-assessed, where individuals were asked how they perceived their income in comparison to a reference group. Relative income calculations using a mean or other summary statistic mirror the seminal papers which first argued that relative income was more important than absolute [4]. In contrast, the self-assessments of relative income identified in the review are a more recent development, and therefore less common, suggesting a shift in our understanding of social comparison. Several different reference groups were used in the studies, and one study [24] found more pronounced relative income effects when comparisons were made to similar individuals (age, education, gender) than to the local population. This suggests that individuals are more concerned with the income of their peers, which aligns with research that found colleagues and friends are the most frequently cited groups when individuals were asked to who they compare their income to [36]. Taking these findings together, it is perhaps more pertinent to use the income of peers or self-assessed relative income when measuring the impact on mental health than to use the average income of the population.

All but one study concluded that there was a relationship between relative income and mental health. This effect was frequently found to be negative (lower relative income compared to reference group was associated with poorer wellbeing). This aligns with a recent meta-review [37] which found that lower subjective social status was associated with poorer mental health.

Income inequality at the regional or national level moderated the relationship between relative income (referred to in the paper in question as subjective social status) and life satisfaction [25], in that in areas with larger inequality, where individuals view themselves in terms of social status becomes more important and so relative status has a larger impact on life satisfaction in areas with high levels of income inequality. This is similar to findings by Schneider [38]. They argue that income inequality mediates the relationship between relative income and wellbeing in that income inequality lowers the self-perception of social status and, in turn, the overall wellbeing of individuals.

Despite the focus on mental health in this review, most of the included studies did not specify mental health disorders and only used subjective wellbeing (SWB) measures such as life satisfaction. Primary research, including validated mental health measures, namely the GHQ-12, were used in one study [24]. Although validated measures would have been preferred, low life satisfaction has been correlated on an individual level with self-reported poor mental health [39] and on a population level, low life satisfaction has been correlated with increased suicide rates and psychiatric hospital admissions [40]. Therefore, life satisfaction still provides a good insight into the relationship between relative income and mental health.

This review focused on individuals in paid employment, and while some studies stratified by gender and age,none compared industries. Given that public sector workers appear to be more dissatisfied with their pay than those in the private sector [41], understanding relative income effects between industries may offer some insight into why such a difference exists. Industry might also be important for the reference group, for example, Frijters et al. [42] found that the higher the expected private sector wage relative to the NHS salary, the more likely nurses were to leave the NHS.

Similarly, whilst important moderating variables such as the reference group were identified, workplace conditions that might help mitigate low relative income were not explored. Pay does not exist in a vacuum and many factors make up workplace experience such as environment, working relationships and workload. Indeed, a systematic review of satisfaction, wages and retention within the NHS [43] found that higher pay would not compensate for other motivations to leave, such as lack of recognition, discrimination and high workload.

## Strengths and limitations

A strength of this review was a rigorous search strategy, co-developed with a data librarian. Nine databases were explored using a broad search strategy, outlined in an a priori PROSPERO-approved review protocol. Additionally, a second, independent reviewer screened and assessed the quality of a proportion of the studies with high inter-rater reliability.

The review does have limitations. As with all systematic reviews, the findings of this review are subject to publication bias. This review did attempt to mitigate this by widening the search to grey literature so that not just peer-reviewed literature was included. However, even grey literature is subject to publication bias and was restricted to 'reputable sources'. Book chapters were excluded as they are often not freely available, but this could have widened the search given that book chapters are a popular output in the field of economics. Additionally, the studies included adjusted for a wide range of moderating variables, and these were included in the narrative synthesis, but it is possible that some were not considered to the same extent as others.

## Implications for policy and practice

The findings of this review have implications for policy and practice. Firstly, the way relative income is defined is pertinent. Except for self-assessed relative income, researchers used mean, median or ratio to split the sample into above/below the reference group income. Notably, the UK government uses median to define relative low pay (or deprivation) and subsequently this division is used to compare the outcomes of those in and out of poverty and develop policies to mitigate these outcomes [31]. As determined in the review, the different measurements and reference group can result in differing effects on satisfaction, either a negative or positive effect [24]. This demonstrates the importance of the measurement method and policymakers should consider the various methods of measuring relative poverty. These should not be mutually exclusive and collectively can help build a better picture of poverty, sentiments echoed by the Chair of the UK Statistics Authority [44].

Income inequality was not the direct focus of this review but was often featured in the multivariate analysis conducted by the included studies. It is clear from this review that income inequality is intrinsically linked to relative income. For instance, Becchetti et al. [25] concluded that aspirational effects are only possible if individuals believe they live in a context of high social mobility. This is particularly relevant in the 'cost of living crisis' where the price of food, rent, mortgage and bills increases quicker than average income rises. Income inequality in the United Kingdom is already relatively large compared to other developed countries [45] and is expected to increase further to record levels (40.8% in 2027–28) [46]. This is partially due to rising interest rates which tend to benefit high earners as they likely have savings and investments which grow with high interest rates [46]. Governments are acutely aware of inflation and policies that successfully manage inflation and bring down living costs could be crucial in alleviating negative relative income effects identified in this review.

Earning less than one's peers was not consistently associated with poor life satisfaction. Different effects were found depending on age, coined the 'tunnel effect'. For individuals under 45/49 years of age, having higher-earning peers was associated with higher satisfaction, possibly because it indicates potential for oneself and something to aspire to [20, 21]. However, not everyone has equal opportunities. Therefore, it is important to address systematic barriers to social progression, such as inequalities in the early development of cognitive, linguistic, and social skills [9].

In addition, barriers to employment also need to be considered to assist with social mobility. The UK government recently released plans to get more people into jobs [47]. However,

2.8 million people in the UK are not working due to long-term sickness [48] and some of these individuals will be on waiting lists for treatment for issues such as mental health or musculoskeletal problems at a time when NHS waiting lists are at a record high [49]. Therefore, initiatives to get individuals back into work need to be combined with policies to tackle waiting lists and foster a healthy workforce.

## Conclusions

In conclusion, this review summarised the literature on relative income and mental health in UK employees and developed a conceptual framework for relative income. Relative income has an effect on mental health and this is most commonly, a negative effect for those who earn less than their reference group, particularly pertinent in a socio-political climate dominated by pay disputes and public sector retention challenges. We also highlighted important moderating factors that can influence this relationship such as gender, age and income inequality, emphasising the importance of policies which aim to close this wealth disparity and tackle poverty. Positive relative income effects identified in this review offer an opportunity for policymakers and private organisations alike to encourage social mobility and create fair and aspirational work environments. Further research is needed on the relationship between relative income and mental disorders, in addition to life satisfaction, to build on the wider wellbeing research. This review only identified one qualitative study. Therefore, more qualitative research to understand whom individuals tend to compare themselves to and the other factors that play a part in employee wellbeing such as work conditions would be beneficial. This review, and future work, contributes to a field which can help to improve the experience and overall health of UK employees.

## Supporting information

**S1-S3 Files. PRISMA checklist, search strategy and list of included papers.**
(DOCX)

**S4 File. Screening spreadsheet.**
(XLSX)

**S5 File. Data extraction spreadsheet.**
(XLSX)

**S6 File. Quality assessment spreadsheet.**
(XLSX)

## Acknowledgments

We would like to thank the expert group who kindly gave their time to review the protocol and for their constructive and helpful feedback on the conceptual framework. We would also like to thank Dr Eirini-Christina Saloniki for her guidance in interpreting the literature and support with the manuscript preparation.

## Author contributions

**Conceptualization:** Bethany Croak, Simon Wessely, Danielle Lamb, Sharon A.M. Stevelink.

**Data curation:** Bethany Croak, Laura E. Grover.

**Formal analysis:** Bethany Croak, Laura E. Grover.

**Funding acquisition:** Simon Wessely, Danielle Lamb, Sharon A.M. Stevelink.

**Supervision:** Simon Wessely, Danielle Lamb, Sharon A.M. Stevelink.

**Validation:** Laura E. Grover, Kalpa Kharicha, Danielle Lamb, Sharon A.M. Stevelink.

**Visualization:** Bethany Croak, Kalpa Kharicha.

**Writing – original draft:** Bethany Croak.

**Writing – review & editing:** Laura E. Grover, Simon Wessely, Kalpa Kharicha, Danielle Lamb, Sharon A.M. Stevelink.

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
