## [Decision Letter · Decision Letter 0]

22 Dec 2024

PONE-D-24-48925Relative income and its relationship with mental health in UK employees: a systematic review.PLOS ONE

Dear Dr. Croak,

Thank you for submitting your manuscript to PLOS ONE. After careful consideration, we feel that it has merit but does not fully meet PLOS ONE’s publication criteria as it currently stands. Therefore, we invite you to submit a revised version of the manuscript that addresses the points raised during the review process.

Please note that we have only been able to secure a single reviewer to assess your manuscript. We are issuing a decision on your manuscript at this point to prevent further delays in the evaluation of your manuscript. Please be aware that the editor who handles your revised manuscript might find it necessary to invite additional reviewers to assess this work once the revised manuscript is submitted. However, we will aim to proceed on the basis of this single review if possible.

We look forward to receiving your revised manuscript.

Kind regards,

Jianhong Zhou

Staff Editor

PLOS ONE

Journal Requirements:

 When submitting your revision, we need you to address these additional requirements. 1. Please ensure that your manuscript meets PLOS ONE's style requirements, including those for file naming. The PLOS ONE style templates can be found at https://journals.plos.org/plosone/s/file?id=wjVg/PLOSOne_formatting_sample_main_body.pdf and https://journals.plos.org/plosone/s/file?id=ba62/PLOSOne_formatting_sample_title_authors_affiliations.pdf. 2. Your abstract cannot contain citations. Please only include citations in the body text of the manuscript, and ensure that they remain in ascending numerical order on first mention. 3. Please include a caption for figure 1. 4. We note that the grant information you provided in the ‘Funding Information’ and ‘Financial Disclosure’ sections do not match.  When you resubmit, please ensure that you provide the correct grant numbers for the awards you received for your study in the ‘Funding Information’ section. 5. Thank you for stating the following financial disclosure:  [This work is part of a PhD nested within the NHS CHECK study and funded by the ESRC [ES/P000703/1]. This report is independent research supported by the National Institute for Health and Care Research ARC North Thames. The views expressed in this publication are those of the author[s] and not necessarily those of the National Institute for Health and Care Research or the Department of Health and Social Care. LEG is funded by the Armed Services Trauma Rehabilitation Outcome [ADVANCE] Charity. Key contributors to this charity are the Headley Court Charity [principal funder], HM Treasury [LIBOR Grant], Help for Heroes, Nuffield Trust for the Forces of the Crown, Forces in Mind Trust, National Lottery Community Fund, Blesma - The Limbless Veterans and the UK Ministry of Defence.].  Please state what role the funders took in the study.  If the funders had no role, please state: ""The funders had no role in study design, data collection and analysis, decision to publish, or preparation of the manuscript."" If this statement is not correct you must amend it as needed. Please include this amended Role of Funder statement in your cover letter; we will change the online submission form on your behalf. 6. Thank you for stating the following in the Competing Interests section: [BC, LEG and DL have no competing interests.  SW is a non-executive director at NHS-England. KK is the deputy director and part funded by the NIHR Policy Research Unit in Health and Social Care Workforce. SAMS is supported by the National Institute for Health and Care Research [NIHR], Maudsley Biomedical Research Centre at South London and Maudsley NHS Foundation Trust and the National Institute for Health and Care Research NIHR Advanced Fellowship [Dr Sharon Stevelink NIHR300592].  The views expressed in this publication are those of the authors and not necessarily those of the ESRC, the NHS or the NIHR.].  Please confirm that this does not alter your adherence to all PLOS ONE policies on sharing data and materials, by including the following statement: ""This does not alter our adherence to  PLOS ONE policies on sharing data and materials.” (as detailed online in our guide for authors http://journals.plos.org/plosone/s/competing-interests).  If there are restrictions on sharing of data and/or materials, please state these. Please note that we cannot proceed with consideration of your article until this information has been declared.  Please include your updated Competing Interests statement in your cover letter; we will change the online submission form on your behalf.  7. Please include captions for your Supporting Information files at the end of your manuscript, and update any in-text citations to match accordingly. Please see our Supporting Information guidelines for more information: http://journals.plos.org/plosone/s/supporting-information. 

Reviewers' comments:

Reviewer's Responses to Questions

**Comments to the Author**

1. Is the manuscript technically sound, and do the data support the conclusions?

Reviewer #1: Yes

2. Has the statistical analysis been performed appropriately and rigorously? 

Reviewer #1: Yes

3. Have the authors made all data underlying the findings in their manuscript fully available?

Reviewer #1: Yes

4. Is the manuscript presented in an intelligible fashion and written in standard English?

Reviewer #1: Yes

5. Review Comments to the Author

Reviewer #1: 1. For introduction, including relevant statistics, such as prevalence rates and correlation coefficients of relative income and mental health would strengthen the argument and provide a more robust foundation for the claims made.

2. For method, please provide the full form of the term 'PRISMA guidelines' and 'PROSPERO' when first mentioned in the text to ensure clarity for readers who may not be familiar with this abbreviation. Next, include the complete keywords and query strings or search strategies used for all nine (9) electronic databases and provide them as an appendix to enhance transparency and allow readers to assess the comprehensiveness and reproducibility of the literature search. Table 1 and Table 2 can be made more compact by minimising the spacing between rows and columns. This will improve readability and reduce unnecessary white space.

3. For results, at least a reference for the Quality Assessment method must be provided and include a thorough explanation of the scoring calculation process to clearly show how the quality of studies was evaluated and scored.

4. For discussion, ensure that the discussion section goes beyond merely stating the findings. Provide a thorough analysis by crystallising the key insights, drawing connections between the results and explaining their implications in the context of existing literature. Additionally, elaborate on how the findings could be applied in practice to address poverty and health issue.

6. PLOS authors have the option to publish the peer review history of their article (what does this mean? ). If published, this will include your full peer review and any attached files.

**Do you want your identity to be public for this peer review?** For information about this choice, including consent withdrawal, please see our Privacy Policy .

Reviewer #1: **Yes: ** ERRNA NADHIRAH BINTI KAMALULIL

---

## [Author Response · Author response to Decision Letter 1]

30 Jan 2025

Please refer to the response to review letter.

---

## [Decision Letter · Decision Letter 1]

13 Feb 2025

PONE-D-24-48925R1Relative income and its relationship with mental health in UK employees: a systematic review.PLOS ONE

Dear Dr. Croak,

Thank you for submitting your manuscript to PLOS ONE. After careful consideration, we feel that it has merit but does not fully meet PLOS ONE’s publication criteria as it currently stands. Therefore, we invite you to submit a revised version of the manuscript that addresses the points raised during the review process.

We look forward to receiving your revised manuscript.

Kind regards,

Nik Ahmad Sufian Burhan

Academic Editor

PLOS ONE

Journal Requirements:

Reviewers' comments:

Reviewer's Responses to Questions

**Comments to the Author**

1. If the authors have adequately addressed your comments raised in a previous round of review and you feel that this manuscript is now acceptable for publication, you may indicate that here to bypass the “Comments to the Author” section, enter your conflict of interest statement in the “Confidential to Editor” section, and submit your "Accept" recommendation.

Reviewer #2: All comments have been addressed

2. Is the manuscript technically sound, and do the data support the conclusions?

Reviewer #2: Yes

3. Has the statistical analysis been performed appropriately and rigorously? 

Reviewer #2: Yes

4. Have the authors made all data underlying the findings in their manuscript fully available?

Reviewer #2: Yes

5. Is the manuscript presented in an intelligible fashion and written in standard English?

Reviewer #2: Yes

6. Review Comments to the Author

Reviewer #2: I have reviewed the original submission and Revision 1. Upon assessing the authors' responses and revisions, I found that Revision 1 has successfully addressed many of the issues I intended to comment on in the original submission in a satisfactory manner. However, there are a few comments on Revision 1 that should be revised to further improve the manuscript:

1. Figure 3 should include wellbeing as part of the outcome.

2. An explanation of moderation should be added to help readers better understand the effect of these moderating factors.

7. PLOS authors have the option to publish the peer review history of their article (what does this mean? ). If published, this will include your full peer review and any attached files.

**Do you want your identity to be public for this peer review?** For information about this choice, including consent withdrawal, please see our Privacy Policy .

Reviewer #2: No

---

## [Editor Report · Decision Letter 2]

19 Feb 2025

Relative income and its relationship with mental health in UK employees: a systematic review.

PONE-D-24-48925R2

Dear Dr. Croak,

We’re pleased to inform you that your manuscript has been judged scientifically suitable for publication and will be formally accepted for publication once it meets all outstanding technical requirements.

Kind regards,

Associate Professor Dr. Nik Ahmad Sufian Burhan

Academic Editor

PLOS ONE

And, 

*Head of Department* , 

Department of Social and Development Sciences, 

Faculty of Human Ecology, 

Universiti Putra Malaysia,

43400 UPM Serdang,

Malaysia. 

Additional Editor Comments (optional):

ACCEPTED FOR PUBLICATION. 
---

## [Editor Report · Acceptance letter]

PONE-D-24-48925R2

PLOS ONE

Dear Dr. Croak,

I'm pleased to inform you that your manuscript has been deemed suitable for publication in PLOS ONE. Congratulations! Your manuscript is now being handed over to our production team.

Kind regards,

on behalf of

Dr. Nik Ahmad Sufian Burhan

Academic Editor

PLOS ONE